# Ginsenoside Rg3 Induces Browning of 3T3-L1 Adipocytes by Activating AMPK Signaling

**DOI:** 10.3390/nu12020427

**Published:** 2020-02-07

**Authors:** Kyungtae Kim, Ki Hong Nam, Sang Ah Yi, Jong Woo Park, Jeung-Whan Han, Jaecheol Lee

**Affiliations:** School of Pharmacy, Sungkyunkwan University, Suwon 16419, Korea; fate514@naver.com (K.K.); nam6422@hanmail.net (K.H.N.); angelna1023@hanmail.net (S.A.Y.); jongwoopark17@gmail.com (J.W.P.); jhhan551@skku.edu (J.-W.H.)

**Keywords:** ginsenoside, Rg3, beige adipocytes, AMPK, browning effect, anti-obesity

## Abstract

Ginsenoside Rg3, one of the major components in *Panax ginseng*, has been reported to possess several therapeutic effects including anti-obesity properties. However, its effect on the browning of mature white adipocytes as well as the underlying mechanism remains poorly understood. In this study, we suggested a novel role of Rg3 in the browning of mature 3T3-L1 adipocytes by upregulating browning-related gene expression. The browning effects of Rg3 on differentiated 3T3-L1 adipocytes were evaluated by analyzing browning-related markers using quantitative PCR, immunoblotting, and immunostaining. In addition, the size and sum area of lipid droplets in differentiated 3T3-L1 adipocytes were measured using Oil-Red-O staining. In mature 3T3-L1 adipocytes, Rg3 dose-dependently induced the expression of browning-related genes such as Ucp1, Prdm16, Pgc1α, Cidea, and Dio2. Moreover, Rg3 induced the expression of beige fat-specific genes (CD137 and TMEM26) and lipid metabolism-associated genes (FASN, SREBP1, and MCAD), which indicated the activation of lipid metabolism by Rg3. We also demonstrated that activation of 5’ adenosine monophosphate-activated protein kinase (AMPK) is required for Rg3-mediated up-regulation of browning gene expression. Moreover, Rg3 inhibited the accumulation of lipid droplets and reduced the droplet size in mature 3T3-L1 adipocytes. Taken together, this study identifies a novel role of Rg3 in browning of white adipocytes, as well as suggesting a potential mechanism of an anti-obesity effect of *Panax ginseng*.

## 1. Introduction

Obesity is medically defined as significantly increased body weight, especially increased portion of white adipose tissue (WAT), which can be associated with several disorders [1]. As obesity is mainly caused by chronically higher food intake than total energy expenditure, maintaining a proper energy balance is important in the treatment of obesity [2]. Currently approved anti-obesity drugs by the U.S. Food and Drug Administration (FDA) inhibit the energy intake either by suppressing the intestinal fat absorption or by repressing appetite; however, they often elicit serious side effects, such as depression, oily bowel movements, and steatorrhea [3]. Alternatively, recent approaches for promoting energy consumption are emerging to treat obesity. Brown adipocytes dissipate stored energy and produce heat via brown fat-specific uncoupling protein 1 (UCP1) [4]. Since the discovery of active brown adipose tissue (BAT) in humans [5], the therapeutic interest concerning browning of white adipose tissue (WAT) to treat obesity [6] and reduce insulin resistance [7] has increased.

In recent studies, inducible brown-like fat cells in WAT, namely, “beige” or “brite” adipocytes have been discovered [8]. Despite having different origins, beige adipocytes (Myf5-negative lineage) and brown adipocytes (Myf5-positive lineage) possess similar morphological and biochemical characteristics, such as the presence of multilocular lipid droplets, enriched mitochondria, and UCP1 expression [9]. Like brown adipocytes, beige adipocytes can dissipate stored energy upon exposure to chronic cold or adrenergic signaling through UCP1-mediated non-shivering thermogenesis. Given the importance of increased energy expenditure in anti-obesity strategy, many studies have been carried out to find endogenous proteins or drugs that induce beige adipocyte in order to establish new therapeutic interventions for obesity [4]. Irisin, an exercise-induced myokine, has been reported to induce beige adipocyte in WAT [10], and inhibition of microRNA-133 (miRNA-133) targeting PR domain containing 16 (PRDM16) has been shown to increase beige adipocyte content in WAT [11]. 

Ginsenoside Rg3 is one of the main active compounds in *Panax ginseng*, which is majorly derived by heat processing of protopanaxadiol (PPD) ginsenosides such as Rb1, Rb2, and Rc from red ginseng (Figure 1A) [12]. Like other ginsenosides, Rg3 has been shown to have multiple therapeutic benefits in antagonizing Adriamycin-induced cardiotoxicity [13], mitigating anxiety in chronic stress [14], and alleviating diabetic complications [15]. Anti-adipogenic effects of Rg3 have also been demonstrated in some studies; however, the effects have been observed in differentiating adipocytes instead of in matured adipocytes [12,16,17].

In this study, we investigated the browning effects of Rg3 in mature 3T3-L1 adipocytes. We observed elevated expressions of BAT-specific and beige adipocyte-specific marker genes in mature 3T3-L1 adipocytes treated with Rg3. The increased expression of WAT-specific and lipid metabolism-related genes were also confirmed in parallel with the accumulation of lipid droplets in mature 3T3-L1 adipocytes. In addition, we found that Rg3 induced browning in 3T3-L1 adipocytes by activating the AMP-activated protein kinase (AMPK) signaling pathways, and AMPK inhibitors alleviated the effects of Rg3. These results suggest that Rg3 may be used as an anti-obesity drug.

## 2. Materials and Methods 

### 2.1. Antibodies and Reagents

Anti-β-actin (Millipore, Temecula, CA; MAB1501), anti-UCP1 (Abcam, Cambridge, MA, USA; ab10983), anti-phospho (T172) AMPKα (Abcam, Cambridge, MA, USA; ab2535), anti-AMPKα (Abcam, Cambridge, MA, USA; ab2532S), anti-Prdm16 (Abcam, Cambridge, MA, USA; ab106410), goat anti-mouse immunoglobulin G (IgG) (Sigma, St. Louis, MO, USA; #AP124P), goat anti-rabbit (Sigma, St. Louis, MO, USA; #401353), anti-4,6-diamidino-2-phenylindole (Sigma, St. Louis, MO, USA), and anti-rabbit antibody Alexa Fluor 594 (Invitrogen, Carlsbad, CA, USA; A11012) were used in this study. Rg3 was purchased from Abcam (Abcam, Cambridge, MA, USA; ab141938).

### 2.2. T3-L1 Cell Culture and Adipogenic Differentiation

Mouse adipocyte-like cell line, 3T3-L1, was purchased from the American Type Culture Collection (ATCC). The cells were maintained in Dulbecco’s modified Eagle’s medium (DMEM) supplemented with 10% bovine calf serum (BCS) and 1% penicillin/streptomycin (P/S). Upon reaching 70%–80% confluency, the cells were allowed to differentiate into adipocytes. For differentiation, the cells were incubated in DMEM with 10% fetal bovine serum (FBS); 1% P/S; and three well-established adipogenic cocktails containing 0.5 mM 3-isobutyl-1-methylxanthine (IBMX), 1 μM dexamethasone, and 1 μg/mL insulin. After 2 days, the medium was replaced with DMEM containing 10% FBS, 1% P/S, and 1 μg/mL insulin every other day.

### 2.3. Protein Extraction and Immunoblotting 

Differentiated 3T3-L1 cells were lysed using Pro-Prep lysis buffer (iNtRON Biotechnology, Korea), and sonicated cell lysates were collected and centrifuged at 13,000 rpm at 4 °C for 15 min. The supernatants were collected and each protein sample was subjected to SDS-polyacrylamide gel electrophoresis (PAGE). Proteins were transferred to polyvinylidene difluoride (PVDF, Millipore, Temecula, CA, USA) membranes using semi-dry transfer (Bio-Rad, Hercules, CA, USA). The membranes were incubated overnight with the indicated primary antibodies, followed by incubation with horseradish peroxidase-conjugated secondary antibodies for 1 h (Millipore, Temecula, CA, USA). The signals were detected using chemiluminescence reagents (Abclon, Korea) and quantified with ImageJ. Every experiment was representative of three independent experiments.

### 2.4. Immunostaining

Differentiated 3T3-L1 cells were fixed using 4% paraformaldehyde in phosphate-buffered saline. Next, the cells were permeabilized using 0.1% Triton X-100 in distilled water and blocked with 1% bovine serum albumin (BSA) in phosphate buffered saline. 3T3-L1 cells were incubated overnight with the indicated primary antibody against uncoupling protein 1 (UCP1) at 4 °C, followed by incubation with the Alexa Fluor-conjugated secondary antibody (Invitrogen, Carlsbad, CA, USA). The nuclei were stained with 4′,6-diamidino-2-phenylindole (DAPI), and the cells were mounted with mounting solution. The signals were detected and analyzed using Cytation 5 (Bio Tek, Winooski, VT, USA).

### 2.5. Quantitative Real-Time PCR (qPCR) 

Total RNA was isolated from the mature cells using Easy-Blue reagent (iNtRON Biotechnology, Korea). Then, the RNA (1 μg) was converted to cDNA using a Maxim RT-PreMix Kit (iNtRON Biotechnology, Korea). Quantitative real-time PCR (qPCR) was accomplished using KAPA SYBR FAST qPCR Master Mix (Kapa Biosystems, Wilmington, MA) and a CFX96 TouchTM real-time PCR detector (Bio-Rad, Hercules, CA). The relative levels of mRNA were normalized to the levels of β-actin mRNA for each reaction. Every experiment was representative of three independent experiments. The primer sequences used for RT-qPCR are as follows: β-actin forward, 5′-ACGGCCAGGTCATCACTATTG-3′; β-actin reverse, 5′-TGGATGCCACAGGATTCCA-3′; Ucp1 forward, 5′-ACTGCCACACCTCCAGTCATT-3′; Ucp1 reverse, 5′-CTTTGCCTCACTCAGGATTGG-3′; Prdm16 forward, 5′-CAGCACGGTGAAGCCATTC-3’; Prdm16 reverse, 5’-GCGTGCATCCGCTTGTG-3′; Pgc1α forward, 5′-CCCTGCCATTGTTAAGACC-3′; Pgc1α reverse, 5′-TGCTGCTGTTCCTGTTTTC-3′; Dio2 forward, 5′-CAGTGTGGTGCACGTCTCCAATC-3′; Dio2 reverse, 5′-TGAACCAAAGTTGACCACCAG-3′; Cidea forward, 5′-TGCTCTTCTGTATCGCCCAGT-3′; Cidea reverse, 5′-GCCGTGTTAAGGAATCTGCTG-3′; Fabp4 forward, 5′-AAGGTGAAGAGCATCATAACCCT-3′; Fabp4 reverse, 5′-TCACGCCTTTCATAACACATTCC-3′; Adipsin forward, 5′-CATGCTCGGCCCTACATG-3′; Adipsin reverse, 5′-CACAGAGTCGTCATCCGTCAC-3′; Adipoq forward, 5′-TGTTCCTCTTAATCCTGCCCA-3′; Adipoq reverse, 5′-CCAACCTGCACAAGTTCCCTT-3′; FASN forward, 5′-TTGACGGCTCACACACCTAC-3′; FASN reverse, 5′-CGATCTTCCAGGCTCTTCAG-3′; SREBP1 forward, 5′-AACGTCACTTCCAGCTAGAC-3′; SREBP1 reverse, 5′-CCACTAAGGTGCCTACAGAGC-3′; MCAD forward, 5′-ACCCTGTGGAGAAGCTGATG-3′; MCAD reverse, 5′-AGCAACAGTGCTTGGAGCTT-3′; CD137 forward, 5′-CCTGTGATAACTGTCAGCCTG-3′; CD137 reverse, 5′-TCTTGAACCTGAAATAGCCTGC-3′; TMEM26 forward, 5′-GCACCATCACTAGAGACCAAC-3′; TMEM26 reverse, 5′-ACAAGAATGCCAGAGACCAG-3′.

### 2.6. Oil-Red-O Staining

The lipid droplets in differentiated 3T3-L1 cells were visualized by Oil-Red-O staining. The matured 3T3-L1 cells were fixed with formalin (10%) for 1 h at room temperature and washed with isopropanol (60%), followed by incubation with Oil-Red-O working solution for 1 h. After that, the cells were rinsed three times with deionized water. For the preparation of Oil-Red-O stock solution, 300 mg of Oil-Red-O powder (Sigma, St. Louis, MO, USA) was dissolved in 100 ml of 99% isopropanol. The Oil-Red-O working solution was a mixture of Oil-Red-O stock solution and water at a 6:4 ratio that was mixed just before use. The stained lipid droplets were captured, and the sum of droplet area was calculated using Gen 5 (Bio Tek, Winooski, VT, USA).

### 2.7. Statistical Analysis

All data are presented as mean ± SEM of at least three independent experiments. Statistical analyses were performed using GraphPad Prism (v.8.0 e). The one-way ANOVA followed by Tukey’s post hoc honestly significant difference (HSD) test was used for multiple comparison correction analysis, and the data were assessed on the basis of the resulting *p*-value (*n.s.*: not significant, * *p* < 0.05, ** *p* < 0.01, *** *p* < 0.001).

## 3. Results

### 3.1. Rg3 Promoted the Expression of Brown and Beige Adipocyte Marker Genes in Mature 3T3-L1 Adipocytes

Firstly, we evaluated the browning effects of Rg3 on mature 3T3-L1 adipocytes. We differentiated 3T3-L1 cells into mature adipocytes using the adipogenic cocktails for 8 days and then treated the cells with different concentrations of Rg3 for 24 h. As Rg3 with a concentration of more than 100 μM is known to have cytotoxic effects on 3T3-L1 adipocytes [12], we treated the cells with 20 and 40 uM of Rg3. Using RT-qPCR analysis, we found that Rg3 treatment dose-dependently increased the mRNA levels of BAT-specific gene markers Ucp1 and Prdm16 (Figure 1B). Consistently, the immunoblotting assay showed that Rg3 up-regulated protein expression of UCP1 and PRDM16 in 3T3-L1 adipocytes (Figure 1C). Furthermore, the immunofluorescence data showed that Rg3 treatment gradually increased the number of UCP1-positive adipocytes (Figure 1D). As the increased expression of Ucp1 and Prdm16 implies the acquisition of brown-like properties, we assessed other BAT-specific marker genes such as Pgc1α, Cidea, and Dio2. Like Ucp1 and Prdm16, Rg3 induced the expression of these marker genes dose-dependently (Figure 1E). The beige adipocytes, which are differentiated from common progenitors with white adipocytes, have been shown to express BAT-specific markers [18]. To test whether Rg3 can induce beige adipocyte-like features in differentiated 3T3-L1 adipocytes, we investigated the mRNA levels of beige adipocyte-specific markers CD137 and TMEM26, which were found to be increased by Rg3 treatment (Figure 1E). These results indicate that Rg3 induces BAT-specific marker genes by up-regulating beige adipocyte-specific marker genes.

### 3.2. Rg3 Reduced white Adipocyte Marker Genes and Increased Lipid Metabolism

We next confirmed the transcriptional regulation of white adipocyte marker genes upon Rg3 treatment. Rg3 treatment in differentiated 3T3-L1 adipocytes for 24 h significantly reduced the expression of white adipocyte marker genes (adipsin, adiponectin, and Fabp4) (Figure 2A). Further, we assessed the mRNA levels of lipogenic marker genes (FASN and SREBP1) and fatty acid β-oxidation marker gene (MCAD). The exposure of 3T3-L1 adipocytes to Rg3 for 24 h increased the expression of both lipogenic (Figure 2B) and fatty acid β-oxidation marker genes (Figure 2C). Rg3 also decreased the accumulation and size of lipid droplets, as observed in Oil-Red-O staining (Figure 2D and Appendix A). These results suggest that Rg3 can decrease the expression of white adipocyte marker genes, as well as up-regulate the lipid metabolism in 3T3-L1 adipocytes.

### 3.3. Rg3 Induced Browning of Differentiated 3T3-L1 Cells Via Activation of AMPK

Previous studies regarding anti-obesity effects of Rg3 have suggested that Rg3 inhibits adipogenic differentiation through AMPK activation [12,16]. We also investigated whether Rg3 activates AMPK signaling in differentiated 3T3-L1 adipocytes and found that AMPK phosphorylation was increased dose-dependently following Rg3 treatment for 24 h (Figure 3A). To assess whether the browning effect of Rg3 on mature 3T3-L1 adipocytes is mediated by activation of the AMPK pathway, we tested the mRNA levels of Ucp1 and Prdm16 using compound C (CC), a commonly used AMPK inhibitor. The data showed that compound C aborted the effect of Rg3 on Ucp1 and Prdm16 (Figure 3B). Consistently, Rg3-induced AMPK phosphorylation and increased protein levels of UCP1 and PRDM16 were reversed by co-treatment with compound C (Figure 3C). These antagonizing effects of compound C were also confirmed by immunostaining using UCP1 antibody (Figure 3D). As Rg3 induced the expression of beige adipocyte-specific marker genes in 3T3-L1 adipocytes, we also tested the expression of TMEM26 and CD137 and found that Rg3-induced increases in expression of these markers was eliminated after compound C co-treatment (Figure 3E). Conclusively, the study findings indicate that the browning effect of Rg3 on differentiated 3T3-L1 adipocytes is mediated by activation of AMPK. 

### 3.4. Rg3 Altered Lipid Metabolism without Affecting white Adipocyte Marker Gene Expression, and the Effects Were Reversed by AMPK Inhibitor

Moreover, we investigated whether Rg3 affects white adipocyte marker genes and lipid metabolism via AMPK. Unlike BAT and beige adipocyte specific marker genes, the co-treatment with compound C and Rg3 additively decreased the expression of white adipocyte marker genes (adipsin, adiponectin and Fabp4) in 3T3-L1 adipocytes (Figure 4A). It found that that Rg3 effects on white adipocyte marker genes were not mediated via AMPK activation. The expression patterns of the lipogenic gene (FASN) and lipolytic gene (MCAD) were similarly altered with that of BAT-specific marker genes (Figure 4B). The Oil-Red-O staining data indicated that Rg3 treatment decreased the accumulation of lipid droplets as well as reduced the droplet size in adipocytes as compared to that in control adipocytes. These effects were reversed after the co-treatment with compound C (Figure 4C and Appendix A). Moreover, the total area of stained lipid droplet was gradually decreased in Rg3-treated 3T3-L1 adipocytes, and compound C alleviated this reduction (Figure 4D). These results indicate that the increased lipid metabolism in Rg3-treated 3T3-L1 adipocytes is mediated via AMPK activation.

## 4. Discussion

Obesity is caused by chronic imbalance between energy intake and energy expenditure [2]. In terms of maintaining proper balance between energy intake and expenditure, dissipating extra energy for thermogenesis by uncoupling protein 1 (UCP1) without diminishing energy intake is an attractive strategy for anti-obesity treatment. Until recently, restraining energy intake by moderating appetite or inhibiting absorption has been considered as a major approach for anti-obesity drugs; however, this practice has often been shown to have side-effects. Therefore, the concept of differentiating a certain portion of white adipocytes into brown-like adipocytes has emerged with great interest. Brown and beige adipocytes have similar characteristics and can spend surplus energy to generate heat by utilizing the thermogenic effector UCP1. In the last decade, several studies have strived to understand the mechanisms of controlling heat production [19], as well as searching the drugs or nutrients to activate these cells. A β3-adrenergic receptor agonist, mirabegron, has been shown to stimulate brown/beige fat and acutely activate human WAT lipolysis [20]. Also, *trans*-cinnamic acid has been found to promote browning of white fat and activate brown adipocytes [21].

An anti-obesity effect of other ginsenosides through browning has also been reported. For instance, it has been found that Rb1 induces browning through the regulation of PPARγ [22], PRDM16, PGC1α, and UCP1 [23] in 3T3-L1 adipocytes. In addition, Rb2 promotes browning of white fat in mice [24], and Rg1 induces browning by upregulating UCP1 expression and mitochondrial activity in white adipocyte [25]. Ginsenoside Rg3 is naturally present at a very low level or trace amounts in American ginseng root extract [26] and Asian ginseng [27]. However, processing to red ginseng and black ginseng significantly increases the Rg3 content in both American ginseng [26] and Asian ginseng [27]. Previous studies on Rg3 effects during 3T3-L1 differentiation have reported that Rg3 attenuates differentiation in 3T3-L1 cells by reducing PPAR-γ, C/EBP-α, FAS, and perilipin [12], or in regulating AMPK and PPAR-γ [16]. In this context, our results suggest the role of Rg3 in browning of mature adipocytes. 

In this study, the treatment with Rg3 resulted in induction of BAT-marker and beige-adipocyte marker genes in mature 3T3-L1 adipocytes dose-dependently (Figure 1E, F). In contrast, WAT-marker genes showed dose-dependent reduction upon treatment with Rg3 for 24 h (Figure 2A). Because 3T3-L1 originates from Myf-5-negative precursor cells, these profiles we observed after treatment with Rg3 can be considered as equivalent to beige adipocytes rather than brown adipocytes. Given the decreased white adipocyte marker gene, we suggest that Rg3 upregulated the activity of beige adipocytes. Moreover, Rg3 treatment caused increased expression of lipid metabolism-associated genes including lipogenesis (SREBP and FASN) and fatty acid β-oxidation (MCAD), which is similar to the finding of chronic β3-adrenergic receptor activation in adipose tissue by CL316,243 [28]. These results are underpinned by decreased accumulation and reduced size of lipid droplets in Rg3-treated 3T3-L1 adipocytes. 

In line with previous studies regarding the effects of Rg3 on AMPK in preadipocytes [16] and myotubes [29], we found that Rg3 increased the phosphorylation of AMPK in mature 3T3-L1 adipocytes. It is well recognized that AMPK is a key modulator of metabolism, which acts by promoting energy producing pathway (catabolism) and suppressing energy storing pathway (anabolism) [30]. AMPK induces WAT browning by elevating PGC1α [31] and PRDM16 [32]. In addition, mTORC1 inhibition by AMPK induces beige adipogenesis [33]. Previous studies confirmed that browning effects of Rb1 and Rg1 also mediated AMPK signaling [23,25] during adipogenesis. In this study, compound C-mediated inhibition of AMPK alleviated the effects of Rg3 on browning and lipid metabolism, indicating that the browning effect of Rg3 is mediated by AMPK activation. Unlike Rb1 and Rg1, AMPK activation by Rg3 promotes browning of mature 3T3-L1. These results suggest that Rg3 could be considered a potential therapeutic candidate for anti-obesity treatment involving induction of white adipocyte browning.

## 5. Conclusion

We provide experimental evidence that ginsenoside Rg3 induces BAT-marker and beige-adipocyte marker genes in mature 3T3-L1 adipocytes dose-dependently. Also, Rg3 represses WAT-marker genes and increases lipid metabolism associated genes. The browning effect of Rg3 on mature 3T3-L1 adipocytes gives rise to decreased accumulation of lipid droplets stained by Oil-red-O. Rg3 increases phosphorylation of AMPK and inhibition with AMPK inhibitor, Compound C, alleviates browning effect of Rg3, but not affect WAT-marker genes. This study suggests that the browning effect of Rg3 is mediated by AMPK activation and Rg3 could be considered of therapeutic candidate for anti-obesity drug.

## Figures and Tables

**Figure 1 nutrients-12-00427-f001:**
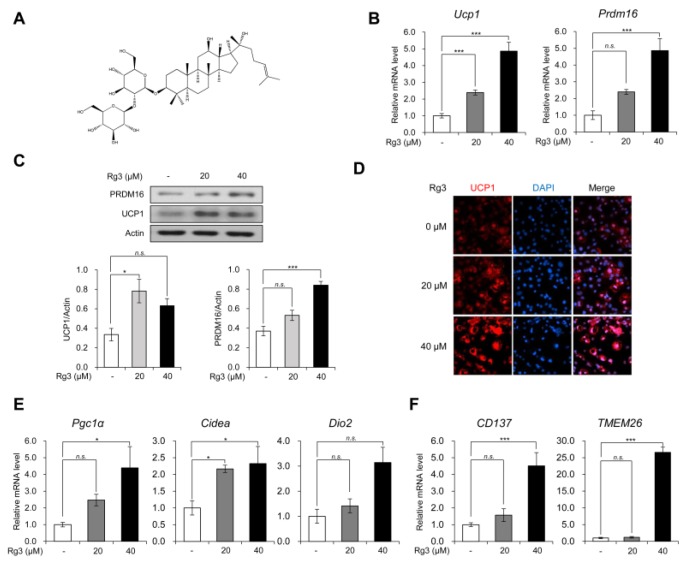
Rg3 promoted the expression of brown and beige adipocyte marker genes in mature 3T3-L1 adipocytes. (**A**) Chemical structure of Rg3. (**B**) The mRNA levels of uncoupling protein 1 (Ucp1) and Prdm16 in differentiated 3T3-L1 cells treated with indicated concentration of Rg3 for 24 h. (**C**) Immunoblot analysis of differentiated 3T3-L1 cells treated with indicated concentration of Rg3 for 24 h (upper) and its quantitative graphs (lower). (**D**) Immunostaining of fully differentiated 3T3-L1 cells treated with indicated concentration of Rg3 for 24 h. (**E**) The mRNA levels of brown adipocyte marker genes in differentiated 3T3-L1 cells treated with indicated concentration of Rg3 for 24 h. (**F**) The mRNA levels of beige adipocyte marker genes in differentiated 3T3-L1 cells treated with indicated concentration of Rg3 for 24 h. Data represent means ± SEM for *n* = 3. Asterisks indicate significant differences from the control (one-way ANOVA; *n.s.*: not significant, * *p* < 0.05, ** *p* < 0.01, *** *p* < 0.001).

**Figure 2 nutrients-12-00427-f002:**
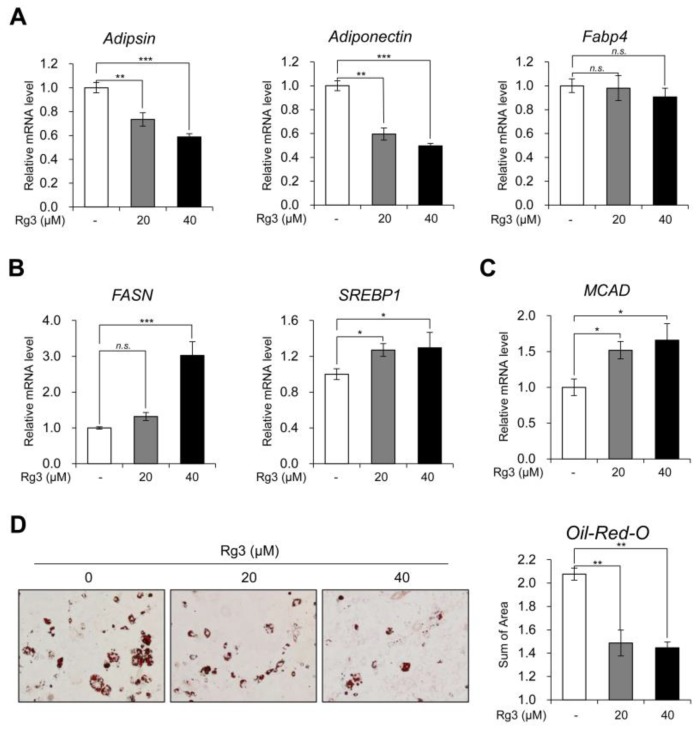
Rg3 reduced white adipocyte marker genes and increased lipid metabolism. (**A**) The mRNA levels of white adipocyte marker genes in differentiated 3T3-L1 cells treated with indicated concentration of Rg3 for 24 h. (**B**) The mRNA levels of lipogenic genes in differentiated 3T3-L1 cells treated with indicated concentration of Rg3 for 24 h. (**C**) The mRNA levels of MCAD in differentiated 3T3-L1 cells treated with indicated concentration of Rg3 for 24 h. (**D**) Oil-Red-O staining showing the accumulation of lipid droplets in differentiated 3T3-L1 cells treated with Rg3 (20, 40 μM) for 24 h (left) and its quantification graph (right) using Gen5 (Bio Tek, Winooski, VT, USA). Data represent means ± SEM for *n* = 3. Asterisks indicate significant differences from the control (one-way ANOVA; *n.s.*: not significant, * *p* < 0.05, ** *p* < 0.01, *** *p* < 0.001).

**Figure 3 nutrients-12-00427-f003:**
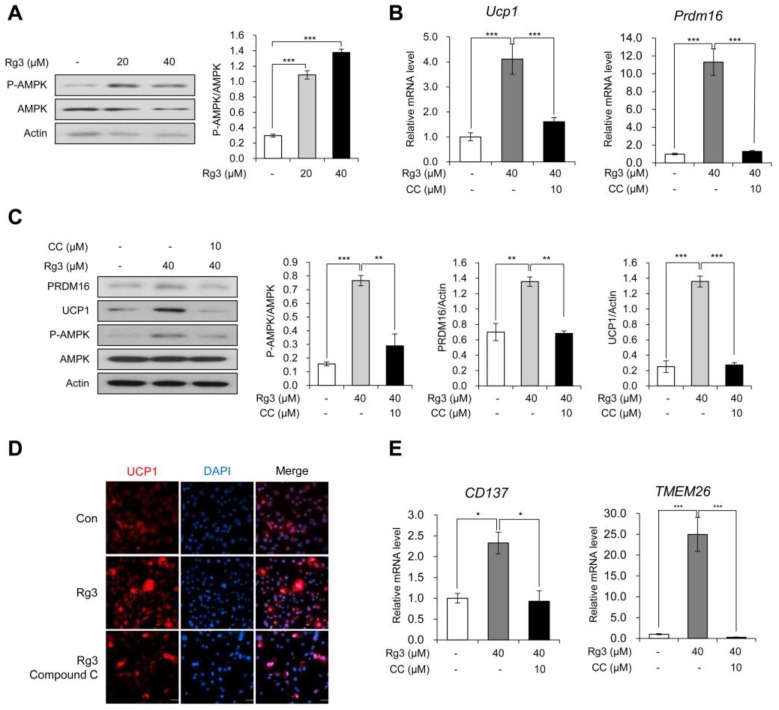
Rg3 induced browning of differentiated 3T3-L1 cells via activation of AMPK. (**A**) Immunoblot analysis of differentiated 3T3-L1 cells treated with indicated concentration of Rg3 for 24 h (left) and its quantitative graphs (right). (**B**) The mRNA levels of Ucp1 and Prdm16 in differentiated 3T3-L1 cells treated with Rg3 (40 μM) in combination with an AMP-activated protein kinase (AMPK) inhibitor, compound C (CC), for 24 h. (**C**) Immunoblot analysis of differentiated 3T3-L1 cells treated with Rg3 (40 μM) in combination with compound C (CC) for 24 h (left) and its quantitative graphs (right). (**D**) Immunostaining of fully differentiated 3T3-L1 cells treated with Rg3 (40 μM), co-treated with or without compound C (CC) for 24 h. (**E**) The mRNA levels of beige adipocyte marker genes in differentiated 3T3-L1 cells treated with Rg3 (40 μM) in combination with compound C (CC) for 24 h. Data represent means ± SEM for *n* = 3. Asterisks indicate significant differences between marked samples (one-way ANOVA; *n.s.*: not significant, * *p* < 0.05, ** *p* < 0.01, *** *p* < 0.001).

**Figure 4 nutrients-12-00427-f004:**
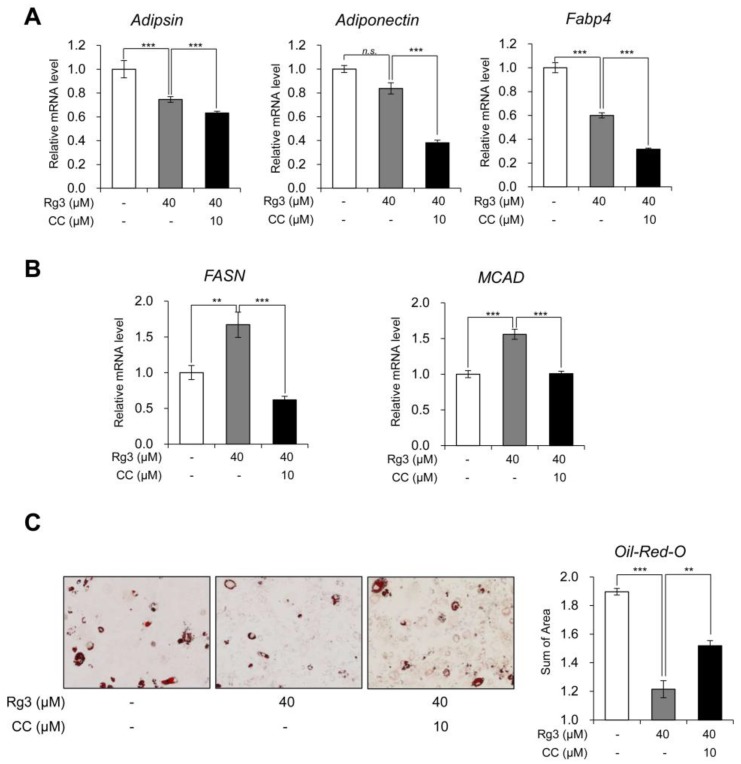
Rg3 altered lipid metabolism without affecting white adipocyte marker gene expression, and the effects were reversed by AMPK inhibitor. (**A**) The mRNA levels of white adipocyte marker genes in differentiated 3T3-L1 cells treated with Rg3 (40 μM) in combination with compound C (CC) for 24 h. (**B**) The mRNA levels of lipogenic genes in differentiated 3T3-L1 cells treated with Rg3 (40 μM) in combination with compound C (CC) for 24 h. (**C**) Oil-Red-O staining showing the accumulation of lipid droplets in differentiated 3T3-L1 cells treated with Rg3 (40 μM) in combination with compound C (CC, 10 uM) for 24 h (left) and its quantitative graph (right) using Gen5 (Bio Tek, Winooski, VT, USA). Data represent means ± SEM for *n* = 3. Asterisks indicate significant differences between marked samples (one-way ANOVA; *n.s.*: not significant, * *p* < 0.05, ** *p* < 0.01, *** *p* < 0.001).

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
