# Peer review of "Ginsenoside Rg3 Induces Browning of 3T3-L1 Adipocytes by Activating AMPK Signaling"

_nutrients, 2020, doi:10.3390/nu12020427_

Round 1

Reviewer 1 Report

The authors clarified the effects of ginsenoide Rg3 on the browing of 3T3-L1 adipocyte through activating AMPK singnaling. In my opinion, your data are interesting, however there are many weak points and there are so many typos.

Point 1: There is typo in title. what is "isnduces"?

Point 2: The authors measured lipid droplet size in 3T3-L1. Therefore, the authors have to denote the method for measureing them and show these results. 

Point 3: In materials and methods section, there is typo on Ln. 78. Waht is Tl-L1 cell?

Point 4: The author should redo the statistical analyses with multiple comparison test for all of data 

Point 5: In Figure 1A, the authors have to show higher resolution diagram.

Point 6: The authors have to more mention and discuss about the adipocyte browning activities of the other ginsenoides and the differences between Rg3 and the others in their activities and results.

Author Response

Response to Reviewer 1 Comments

We thank to the reviewer for recognizing the potential importance of our findings. We feel our responses to their concerns have greatly improved the manuscript. Our responses (red words) to the reviewer’s queries (black words) are described point-by-point below.

(Reviewer’s comments)

The authors clarified the effects of ginsenoide Rg3 on the browing of 3T3-L1 adipocyte through activating AMPK singnaling. In my opinion, your data are interesting, however there are many weak points and there are so many typos.

Point 1: There is typo in title. what is "isnduces"?

Response 1: We corrected the typo (page 1, line 2). Thanks to figure out our mistakes.

Point 2: The authors measured lipid droplet size in 3T3-L1. Therefore, the authors have to denote the method for measureing them and show these results.

Response 2: The data for lipid droplet size in 3T-L1 was attached at supplementary figures. But we didn’t measure the size of lipid droplet. Instead of that, we calculated the sum of droplet area using the software Gen5 (BioTek).

Point 3: In materials and methods section, there is typo on Ln. 78. Waht is Tl-L1 cell?

Response 3: We corrected the typo. Thanks to figure out our mistakes.

Point 4: The author should redo the statistical analyses with multiple comparison test for all of data

Response 4: We clarified the compared group for student’s t test (Page 5, line 183-184; Page 6, line 204-205; Page 6, line 216-217; Page 7, line 242-243). As regards justification of selected statistical analysis, student’s t test is the most popular parametric test for handling statistic data. Since we compared dCt (Ct[ref]-Ct[goi]) for qPCR or quantified value for immunoblot between two groups, it is valid to use t-test.

Point 5: In Figure 1A, the authors have to show higher resolution diagram.

Response 6: Thanks to your point, we uploaded the figures modified to higher resolution (300dpi).

Point 6: The authors have to more mention and discuss about the adipocyte browning activities of the other ginsenoides and the differences between Rg3 and the others in their activities and results.

Response 6: We added discuss about the adipocyte browning activities of the other ginsenosides (Rb1, Rg1) and difference between Rg3 and the others. (Page 8, Line 273-277, Page9, Line 300-302, 304)

Reviewer 2 Report

The article submitted by Kyungtae Kim and collaborators is about the potential role of Ginsenoside Rg3, an  active compound found in Panax ginseng,  in the shift from white to beige adipose tissue phenotype. The study is globally well described, but has some limitations that are described in the present report. Several similar studies were carried out on similar extracts from Ginseng since more than 15 compounds could be isolated. A commercial form was used for the present study. In comparison to previous studies on this family of molecules and adipogenesis, the originality of the present work is the use of differentiated adipocytes. However, the differentiation state appear incomplete in the control group (% of fully differentiated adipocytes is low). The use of the molecule in an animal model or with primary cells would be a plus for this article.  

2.2.3   .3 T3-L1 cell culture and adipogenic differentiation

The protocol for cell differentiation is not the most commonly found in the literature. The differentiation cocktail is usually maintained for 2-3 days before incubation in the presence of insulin for 2 additional days and with FBS only after.

Lipid accumulation was moderate according to the images provided by the authors.

Only 3 replicates were used for each experiments, it Should be mentioned if it is 3 independent experiments that were conducted or if it is 3 replicates from the same batch of cells that were thawed.

We can understand that groups treated with Rg3 were compared to the control group (using student t test) in Fig 1 and 2, but it should be clearly mentioned. In some cases an ANOVA would have been more appropriated, but the results that are illustrated seem convincing.

Fig 2B, significance for FASN at 40µM is missing ?

Author Response

Response to Reviewer 2 Comments

We thank to the reviewer for recognizing the potential importance of our findings. We feel our responses to their concerns have greatly improved the manuscript. Our responses (red words) to the reviewer’s queries (black words) are described point-by-point below.

(Reveiwer’s comments)

The article submitted by Kyungtae Kim and collaborators is about the potential role of Ginsenoside Rg3, an active compound found in Panax ginseng, in the shift from white to beige adipose tissue phenotype. The study is globally well described, but has some limitations that are described in the present report. Several similar studies were carried out on similar extracts from Ginseng since more than 15 compounds could be isolated. A commercial form was used for the present study. In comparison to previous studies on this family of molecules and adipogenesis, the originality of the present work is the use of differentiated adipocytes. However, the differentiation state appear incomplete in the control group (% of fully differentiated adipocytes is low). The use of the molecule in an animal model or with primary cells would be a plus for this article.

Point 1: 2.2.3 3T3-L1 cell culture and adipogenic differentiation. The protocol for cell differentiation is not the most commonly found in the literature. The differentiation cocktail is usually maintained for 2-3 days before incubation in the presence of insulin for 2 additional days and with FBS only after.

Response 1: We used the chemically-induced 3T3-L1 differentiation protocol provided by ATCC. And this protocol is also usually used (For example, Ryu et al., 2018, Science). We used the differentiation cocktail for 2 days and corrected the protocol to clarify. (Page 2, Line 87)

Point 2: Lipid accumulation was moderate according to the images provided by the authors..

Response 2: To reduce batch difference between wells, the differentiated 3T3-L1 were replated before Rg3 treatment as described in Kajimoto et al., 2012. Although lipid accumulation appears to be moderate, we chose a method with less batch difference.

Point 3: Only 3 replicates were used for each experiments, it Should be mentioned if it is 3 independent experiments that were conducted or if it is 3 replicates from the same batch of cells that were thawed.

Response 3: We added the information about replicates in ‘Material and Methods’ (Page 3, Line 99; Page 3, Line 116-117).

Point 4: We can understand that groups treated with Rg3 were compared to the control group (using student t test) in Fig 1 and 2, but it should be clearly mentioned. In some cases an ANOVA would have been more appropriated, but the results that are illustrated seem convincing.

Response 4: We clarified the compared group for student’s t test (Page 5, line 183-184; Page 6, line 204-205; Page 6, line 216-217; Page 7, line 242-243). As regards justification of selected statistical analysis, student’s t test is the most popular parametric test for handling statistic data. Since we compared dCt (Ct[ref]-Ct[goi]) for qPCR or quantified value for immunoblot between two groups, it is valid to use t-test.

Point 5: Fig 2B, significance for FASN at 40µM is missing ?

Response 6: Thanks to your comment, we corrected significance for FASN in figure 2B.

Round 2

Reviewer 1 Report

The authors have to reconsider the statistical analysis.

T-test can only compare two groups at a time . It means that it is generally consider "illegal" to use t-test over and over again on different groups from a single experiment.

Author Response

Reviewer #1

The authors have to reconsider the statistical analysis.

T-test can only compare two groups at a time . It means that it is generally consider "illegal" to use t-test over and over again on different groups from a single experiment

- We changed statistical analyses in all figures using one-way ANOVA as reviewer pointed out and clarified in methods (Page 4, Lin 149-152; Page 5, Line 184; Page 6, Line 205-206; Page 6, Line 218; Page 7, Ling 244). Statistical analyses were performed using GraphPad Prism (v.8.0e). The one-way ANOVA followed by Tukey’s post hoc HSD test was used for multiple comparison correction analysis.

Reviewer 2 Report

the comments were considered and answers were provided.

Author Response

Reviewer #2

the comments were considered and answers were provided.

- We greatly thanks to review our manuscript.